# Lenvatinib or Sorafenib Treatment Causing a Decrease in Skeletal Muscle Mass, an Independent Prognostic Factor in Hepatocellular Carcinoma: A Survival Analysis Using Time-Varying Covariates

**DOI:** 10.3390/cancers15174223

**Published:** 2023-08-23

**Authors:** Kenji Imai, Koji Takai, Shinji Unome, Takao Miwa, Tatsunori Hanai, Atsushi Suetsugu, Masahito Shimizu

**Affiliations:** Department of Gastroenterology/Internal Medicine, Graduate School of Medicine, Gifu University, 1-1 Yanagido, Gifu 501-1194, Japan; takai.koji.t2@f.gifu-u.ac.jp (K.T.); unome-shinji@hotmail.com (S.U.); miwa.takao.a6@f.gifu-u.ac.jp (T.M.); hanai.tatsunori.p8@f.gifu-u.ac.jp (T.H.); suetsugu.atsushi.e2@f.gifu-u.ac.jp (A.S.); shimizu.masahito.j1@f.gifu-u.ac.jp (M.S.)

**Keywords:** hepatocellular carcinoma, lenvatinib, sorafenib, prognostic factor, sarcopenia

## Abstract

**Simple Summary:**

Skeletal muscle depletion is one of the established prognostic factors for hepatocellular carcinoma (HCC). This study clearly demonstrated that skeletal muscle mass continued to decrease significantly during lenvatinib (LEN) or sorafenib (SOR) treatment, which has recently played a major role in the treatment of unresectable HCC. Furthermore, the survival analysis using time-varying covariates in this study revealed that it was one of the independent prognostic factors together with tumor makers and liver functional reserve. These results can help improve the management of unresectable HCC because they suggest that it is essential to prevent skeletal muscle depletion, especially in using LEN/SOR to improve survival in HCC patients.

**Abstract:**

This study aimed to assess the effects of lenvatinib (LEN) or sorafenib (SOR) treatment for hepatocellular carcinoma (HCC) on body composition and changes in body composition on survival. This study enrolled 77 HCC patients. Skeletal muscle index (SMI), subcutaneous and visceral adipose tissue indices (SATI and VATI), AFP, PIVKA-II, and ALBI scores were analyzed at the time of LEN/SOR introduction, three months after the introduction, at treatment discontinuation, and the last observational time. The differences between chronological changes in these values were analyzed using a paired *t*-test. The Cox proportional hazards model was used to analyze prognostic factors using time-varying covariates. The chronological changes in each factor were 45.5–43.6–40.6–39.8 (cm^2^/m^2^) for SMI, 41.7–41.6–36.3–33.7 (cm^2^/m^2^) for SATI, 41.9–41.1–37.1–34.8 (cm^2^/m^2^) for VATI, 2.379–26.42–33.61–36.32 (×10^3^ ng/mL) for AFP, 9.404–13.39–61.34–25.70 (×10^3^ mAU/mL) for PIVKA-II, and −2.56–−2.38–−1.99–−1.90 for the ALBI score. The presence of pre-treatment (*p* = 0.042), AFP (*p* = 0.002), PIVKA-II (*p* < 0.001), ALBI score (*p* < 0.001), and SMI (*p* = 0.001) were independent prognostic factors. Skeletal muscle mass decreases significantly during LEN/SOR treatment and is an independent prognostic factor for HCC.

## 1. Introduction

Hepatocellular carcinoma (HCC) is a malignancy with high morbidity and mortality [1,2] which often develops in patients with chronic hepatitis or cirrhosis that was caused by a chronic infection with hepatitis B or C viruses, heavy alcohol consumption, being overweight or obese, or other diseases [2,3]. Both liver function impairment and tumor factors such as large tumor foci, the multiplicity of tumors, pathologically high-grade atypia of tumor cells, and the presence of portal venous invasion negatively affect the survival of patients with HCC [2,4]. In recent years, it has also been reported that skeletal muscle depletion and sarcopenia are prognostic factors for patients with various malignant diseases, including HCC. For instance, skeletal muscle depletion at the start of treatment has worsened survival in HCC patients at all tumor stages and in those treated with sorafenib (SOR) [5,6].

SOR is the first oral active multikinase inhibitor that has been confirmed to prolong survival in patients with unresectable HCC [7]. Since then, other multikinase inhibitors and immune checkpoint inhibitors, including lenvatinib (LEN), regorafenib, cabozantinib, and atezolizumab plus bevacizumab, have been used as anticancer drugs for unresectable HCC [8,9,10,11]. Systemic therapy using these drugs is expected to demonstrate an antitumor effect and has recently played a major role in the treatment of this malignancy [12,13,14]. However, these drugs can often cause various gastrointestinal adverse events (AEs), including anorexia, diarrhea, nausea, and vomiting, which lead to body weight loss and malnutrition. For example, the REFLECT trial reported that 22% of the patients receiving SOR and 31% of those receiving LEN experienced weight loss [8]. Therefore, weight loss and its associated changes in body composition are common AEs that occur during the systemic therapy of patients with advanced HCC.

Interestingly, in addition to skeletal muscle depletion at the start of treatment [5,6], rapid depletion of skeletal muscle mass and subcutaneous adipose tissue mass after the induction of SOR has been reported to be associated with the poor survival of patients with HCC [6,15]. These facts suggest that systemic therapy for HCC can cause skeletal muscle and adipose tissue depletion and these drug-induced unfavorable changes in body composition negatively affect survival. However, no studies have evaluated the chronological changes in body composition after the introduction of systemic therapy for HCC by periodic computed tomography (CT) examinations and how these changes affect clinical outcomes, such as survival rates.

In this study, we periodically measured the chronological changes in body composition as well as the established prognostic factors, such as tumor markers and liver functional reserve, from the introduction of LEN/SOR treatment to the last observational period. Furthermore, we conducted a survival analysis to clarify whether changes in body composition, together with other possible prognostic factors, would affect survival after the introduction of LEN/SOR treatment. We treated all the covariates in this study as time-varying covariates because body composition, tumor markers, and liver functional reserve may change their values over time [16].

## 2. Materials and Methods

### 2.1. Patients, Treatment, and a Follow-Up Strategy

A total of 109 HCC patients were treated with LEN/SOR between May 2009 and December 2021 at the Gifu University Hospital. Among them, the 77 patients who continued to receive LEN/SOR treatment for at least 3 months and were followed by periodic CT examinations were enrolled. Decisions pertaining to the LEN/SOR treatment were made according to the Clinical Practice Guidelines for HCC that were issued by the Japan Society of Hepatology [17]. Each patient’s therapeutic response was judged using dynamic CT according to the Response Evaluation Criteria in Cancer of the Liver [18]. This is considered an appropriate system for the assessment of the post therapeutic response of HCC to SOR [19]. AEs were assessed according to the Common Terminology Criteria for Adverse Events, version 5.0.

Overall survival (OS) and progression free survival (PFS) were defined as the interval that started with the date of the introduction of LEN/SOR and ended with the date of death or progressive disease (PD). If these events did not happen, they were defined to the last observational day. When PD, the deterioration of liver functional reserve or severe AEs were observed, so a change in the treatment strategies for HCC was considered. All study participants provided verbal informed consent. Moreover, we provided the participants with an opportunity to opt out by disclosing the details of the study. The study design, including this consent procedure, was approved by the ethics committee of the Gifu University School of Medicine (ethical protocol code: 29–26).

### 2.2. Measurement of the Chronological Changes in Body Composition, Tumor Makers, and Liver Functional Reserve

The CT cross-sectional areas (cm^2^) of the volume of skeletal muscle and subcutaneous and visceral adipose tissues were measured using SYNAPSE VINCENT software (version 6.7, Fujifilm Medical, Tokyo, Japan). The obtained values were normalized by the square of the patient’s height (m^2^) to obtain the skeletal muscle index (SMI, cm^2^/m^2^), subcutaneous adipose tissue index (SATI, cm^2^/m^2^), and visceral adipose tissue index (VATI, cm^2^/m^2^) [20]. Data on SMI, SATI, VATI, and alpha-fetoprotein (AFP) proteins induced by vitamin K absence or antagonist-II (PIVKA-II), as well as the ALBI score, which represented liver functional reserve, were collected at the time of LEN/SOR introduction, three months after the introduction, at drug discontinuation, and the last observational time.

### 2.3. Statistical Analysis

OS and PFS were estimated using the Kaplan–Meier method and the differences between the curves were evaluated using a log-rank test. Differences between chronological changes in the SMI, SATI, VATI, AFP, PIVKA-II, and ALBI scores were analyzed using the Wilcoxon paired test. Overall *p* values for the time series of those parameters were also analyzed using a mixed-effects linear model for repeated measures. The Cox proportional hazards model was used to determine which of the factors described above affected OS. When the survival analyses in this study were carried out, all the factors were treated as time-varying covariates [16]. Statistical significance was defined as *p* < 0.05. All statistical analyses were performed using R software ver. 4.2.2 (R Foundation for Statistical Computing, Vienna, Austria; http://www.R-project.org/ (accessed on 1 February 2023).

## 3. Results

### 3.1. Baseline Clinical Characteristics and Treatment Course of the Enrolled Patients

The baseline clinical characteristics of the enrolled patients (66 men and 11 women; median age, 73 years) just before the introduction of LEN/SOR are presented in Table 1. Among the enrolled patients, 28 were on LEN and 49 on SOR. The patient’s treatment course is listed in Table 2. A total of 71 patients (92.2%) had undergone treatment before beginning LEN/SOR, 49 (63.6%) received combination treatment, mainly transcatheter arterial chemoembolization or radiation therapy during LEN/SOR treatment, and 31 (40.3%) received further treatments after discontinuing LEN/SOR.

### 3.2. Therapeutic Effect and Adverse Events of the Enrolled Patients

The therapeutic effect of the complete response (CR), partial response (PR), stable disease (SD), and PD was 7, 15, 20, and 35 cases, respectively (Table 1). The objective response and disease control rates were 28.6% and 54.5%, respectively. OS rates at 1, 3, and 5 years and median OS were 81.1%, 26.2%, 12.0%, and 23.2 months, respectively (Figure 1a), whereas the PFS rates at 1, 3, and 5 years and median PFS were 55.4%, 11.5%, 5.9%, and 14.5 months, respectively (Figure 1b). During the observation period, 68 patients experienced PD and 53 died in the present study. The median follow-up periods for all patients, those treated with LEN and those with SOR, were 20.9, 15.6, and 21.8 months. There were no significant differences in OS (*p* = 0.922) and PFS (*p* = 0.916) between LEN and SOR groups (Figure 1c,d).

Table 3 lists the AEs recorded in response to the LEN/SOR treatment. Overall, 75 patients (97.4%) experienced some form of AEs, the most frequent of which at any grade was appetite loss (58.4%) followed by hand–foot syndrome (46.8%) and general fatigue (40.3%). AEs at grade ≥ 3 were identified in 24 patients (31.2%), the most frequent of which was hypertension (10.4%) followed by proteinuria (5.2%) and hemorrhage (5.2%).

### 3.3. Chronological Changes in Body Composition, Tumor Makers, and Liver Functional Reserve

Chronological changes in body composition at the time of LEN/SOR introduction, three months after the introduction, at drug discontinuation, and the last observational time were 45.5–43.6–40.6–39.8 (cm^2^/m^2^) for SMI, 41.7–41.6–36.3–33.7 (cm^2^/m^2^) for SATI, and 41.9–41.1–37.1–34.8 (cm^2^/m^2^) for VATI, respectively (Figure 2a–c). SMI significantly diminished within three months after the introduction (*p* < 0.01) and from then to the discontinuation (*p* < 0.01). Both SATI and VATI did not decrease during the first three months of treatment but significantly decreased thereafter (*p* < 0.01 and *p* < 0.01). A similar trend was observed when the LEN and SOR groups were analyzed separately (Figure 2d–f).

Chronological changes in AFP and PIVKA-II were 2.379–26.42–33.61–36.32 (×10^3^ ng/mL) and 9.404–13.39–61.34–25.70 (×10^3^ mAU/mL), respectively (Figure 3a,b). Both values significantly increased from three months after the introduction to the discontinuation (*p* < 0.01 and *p* < 0.01). Those in the ALBI score were −2.56, −2.38, −1.99, and −1.90 which significantly continued to deteriorate over time (*p* < 0.01, *p* < 0.01, and *p* = 0.01, Figure 3c). Overall, *p* values for time series for SMI, SATI, VATI, AFP, PIVKA-II, and ALBI scores were <0.001, <0.001, <0.001, 0.286, <0.001, and <0.001, respectively.

### 3.4. Analysis of the Factors Affecting Survival Using Time-Varying Covariates in the Cox Proportional Hazards Model

Table 4 presents the results of the multivariate analysis performed to determine the factors affecting OS using time-varying covariates in the Cox proportional hazards model. The results demonstrated that the presence of pretreatment (hazard ratio [HR]: 2.995, 95% confidence interval [CI]: 1.040–8.620, *p* = 0.042), AFP (HR: 1.002, 95% CI: 1.001–1.002, *p* = 0.002), PIVKA-II (HR, 1.004; 95% CI, 1.002–1.006; *p* < 0.001), ALBI score (HR: 3.609, 95% CI: 2.342–5.562, *p* < 0.001), and SMI (HR: 0.941, 95% CI: 0.906–0.977, *p* = 0.001) were independent prognostic factors for advanced HCC patients treated with LEN/SOR. We also evaluated factors that could affect PFS in the same manner (Appendix A). AFP (HR: 1.002, 95% CI: 1.001–1.003, *p* = 0.007), PIVKA-II (HR: 1.004, 95% CI: 1.001–1.008, *p* = 0.012), and the ALBI score (HR: 1.676, 95% CI: 1.145–2.452, *p* = 0.008) were independent factors that would affect PFS.

To clarify the clinical characteristics of those who experienced rapid depletion of skeletal muscle, we defined a total of 19 patients who experienced > 20% depletion of SMI during LEN/SOR treatment as a rapid SMI depletion group and compared them with those in the non-SMI depletion group (Appendix A). Patients in the rapid SMI depletion group had significantly shorter survival than those in the non-SMI depletion group (15.7 vs. 23.1 months, *p* < 0.001). However, there were no differences between the two groups concerning age, sex, therapeutic effect, drugs, the presence of combination treatment, liver functional reserve, clinical tumor stage at the introduction, and AEs.

## 4. Discussion

The results of this study clearly demonstrated that skeletal muscle mass continued to decrease significantly during LEN/SOR treatment in patients with advanced HCC and that it was one of the independent prognostic factors. To the best of our knowledge, this was the first study that attempted to identify the prognostic factors in HCC patients treated with LEN/SOR using time-varying covariates in a Cox proportional hazards model. We think that it is more appropriate to analyze the data using time-varying covariates than using only one baseline value, which is the conventional way. In this case, covariates, such as body composition, tumor makers, and liver functional reserve, drastically change their value over time, as demonstrated in Figure 2 and Figure 3.

There are several possible causes that can make the volume of skeletal muscle decrease during LEN/SOR treatment. First, LEN and SOR may decrease skeletal muscle synthesis by directly inhibiting the signaling of the growth factors involved in protein synthesis, such as vascular endothelial growth factor receptors [21]. Second, treatment-related AEs may promote the depletion of skeletal muscle. In this study, 58.4% and 40.3% of the enrolled patients experienced appetite loss and general fatigue, respectively. Patients who have reduced nutritional intake and decreased activity because of AEs tend to have a decrease in their skeletal muscle. Third, HCC and liver cirrhosis itself may promote the depletion of skeletal muscle because they are both major causes of secondary sarcopenia [22,23]. Against our expectations, there were no differences in drugs, AEs, liver functional reserve, clinical tumor stage, and therapeutic effect between the rapid SMI depletion and control groups in this study. These findings might imply that there are other unknown mechanisms that associate LEN/SOR treatment with the depletion of skeletal muscle.

Sarcopenia, a progressive and generalized skeletal muscle disorder characterized by the loss of skeletal muscle mass and strength, is an important complication of chronic liver disease [23]. Sarcopenia worsens survival in patients with cirrhosis and HCC [5,24]. Cirrhotic patients with low hepatic functional reserves show a rapid decrease in skeletal muscle mass which determines their prognosis [25]. These reports and the results of this study, which revealed that LEN/SOR treatment decreases skeletal muscle mass over time, in unresectable with regard to HCC, again suggesting that patients with cirrhosis and HCC tend to lose skeletal muscle mass, which is a prognostic factor, as the underlying liver diseases progress. In recent years, advances in multidisciplinary treatment have made post-progression survival more important in HCC therapy [26]. Therefore, it is noteworthy that skeletal muscle depletion has been shown to significantly worsen post-progression survival, especially in patients who have undergone SOR or LEN [27,28].

To prevent the depletion of skeletal muscle and to accordingly improve survival in HCC patients treated with LEN/SOR, it is of the most importance to maintain liver functional reserve and reduce tumor burden to the best extent possible. To achieve this goal, LEN/SOR treatment must be supplemented by practicing proper nutrition and exercise therapies and by properly managing AEs. Administration of branched-chain amino acids (BCAAs), which are the basic nutritional therapy in liver disease patients with sarcopenia, may be a valuable therapy for improving the prognosis of HCC patients treated with SOR [29,30,31]. Exercise therapy has increased skeletal muscle mass, which is associated with an improvement in the prognosis in HCC patients [32]. The combination of BCAA administration and exercise therapy has also been reported to minimize skeletal muscle atrophy in patients with HCC [33]. In addition, proper nutrition and exercise therapies are important in combating obesity and diabetes, which increase the risk of HCC and worsen prognosis [34]. Insulin resistance and increased visceral fat volume, which may be targeted for nutritional and exercise therapies, are risk factors for HCC recurrence [20,35]. On the other hand, in addition to performing nutrition and exercise therapies, if AEs leading to skeletal muscle depletion do not improve, it is necessary to immediately consider reducing the dose of LEN/SOR and changing the treatment strategy for HCC.

In this study, skeletal muscle mass decreased rapidly from the time of treatment induction and that predicted prognosis while adipose tissue mass was relatively preserved from the start of treatment and was not a predictor. In contrast, it has been reported that rapid depletion of SATI during SOR treatment is significantly associated with poor survival in patients with HCC [15]. Since the SAT functions as a metabolic reservoir to store excess energy [36], it may be the primary source of energy when treatment induces a negative energy balance. Further studies are needed to determine the prognostic impact of the reduction in SATI, which may be an indicator of extreme nutritional impairment, on patients with HCC. It should also be noted that both SATI and VATI continue to decline after the end of chemotherapy (Figure 2b,c), probably related to cachexia and end-stage nutritional disturbances.

This study has several limitations. First, this was a retrospective single-center study with a comparatively small sample size. Second, 63.6% of the enrolled patients in this study received other combination treatments during LEN/SOR treatment. Thus, we could not estimate the pure effects of LEN/SOR on body composition, liver functional reserve, and therapeutic response. Third, OS may have been influenced not only by the LEN/SOR treatment but also by concomitant treatments and post treatments. These limitations mainly resulted from the retrospective research method used in this study. To clarify the effects of LEN/SOR treatment on body composition and the changes in body composition observed in this study on OS more persuasively, a prospective study involving a larger number of patients enrolled from several centers should be performed.

## 5. Conclusions

Skeletal muscle mass in HCC patients continued to significantly decrease during treatment with LEN/SOR and this was found to be one of the independent prognostic factors along with the established prognostic factors, such as AFP, PIVKA-II, and the ALBI score, using time-varying covariates in the Cox proportional hazards model.

## Figures and Tables

**Figure 1 cancers-15-04223-f001:**
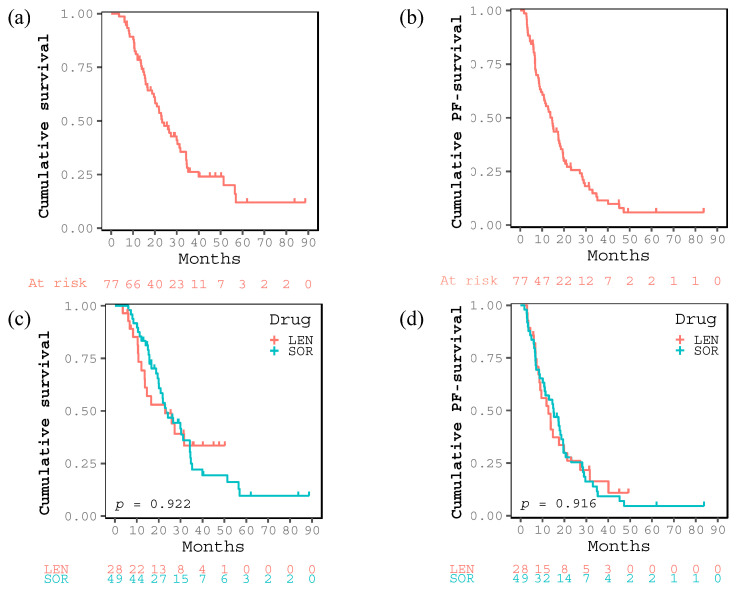
Kaplan–Meier curves for (**a**) cumulative survival and (**b**) cumulative progression free (PF) survival after introducing sorafenib or lenvatinib of all the enrolled patients. Kaplan–Meier curves for (**c**) cumulative survival and (**d**) cumulative PF-survival in the two groups treated with lenvatinib or sorafenib.

**Figure 2 cancers-15-04223-f002:**
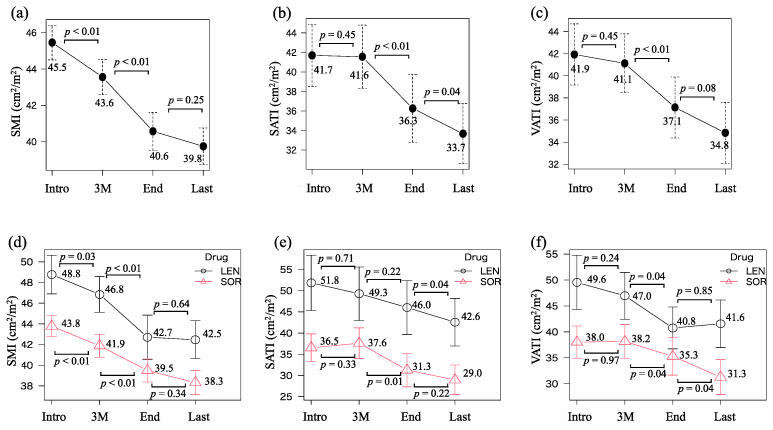
Chronological changes in (**a**) skeletal muscle index (SMI), (**b**) subcutaneous adipose tissue index (SATI), and (**c**) visceral adipose tissue index (VATI) at the introduction (Intro), three months after the introduction (3 M), at the discontinuation of lenvatinib or sorafenib treatment (end), and the last observational time (Last). The values in (**d**) SMI, (**e**) SATI, and (**f**) VATI were divided into two groups that were treated with lenvatinib or sorafenib.

**Figure 3 cancers-15-04223-f003:**
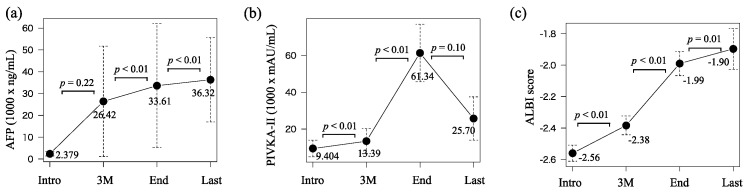
Chronological changes in (**a**) alpha-fetoprotein (AFP), (**b**) proteins induced by vitamin K absence or antagonist-II (PIVKA-II), and (**c**) ALBI score at the introduction (intro), three months after the introduction (3 M), at the discontinuation of lenvatinib or sorafenib treatment (end), and the last observational time (Last).

**Table 1 cancers-15-04223-t001:** Baseline demographic and clinical characteristics of the enrolled patients.

Variables	All Patients (*n* = 77)
Age (years)	73 (65–78)
Sex (male/female)	66/11
Etiology (HBV/HCV/others)	16/33/28
Drug (Lenvatinib/Sorafenib)	28/49
BCLC stage (B1/B2/C)	12/24/41
SMI (cm^2^/m^2^)	44.3 (40.7–49.3)
SATI (cm^2^/m^2^)	38.1 (22.7–48.1)
VATI (cm^2^/m^2^)	40.7 (21.1–54.2)
Child Pugh score (5/6/7/8/9)	59/16/1/0/1
ALBI score	−2.62 (−2.91–−2.23)
ALB (g/dL)	3.9 (3.6–4.3)
AST (U/L)	33 (25–44)
ALT (U/L)	23 (17–32)
T-Bil (mg/dL)	0.8 (0.7–1.0)
PT (%)	93 (84–102)
AFP (ng/mL)	48.8 (8.5–431.7)
PIVKA-II (×10^3^ mAU/mL	291 (32–1679)
Best response (CR/PR/SD/PD)	7/15/20/35

Continuous covariates are presented as medians (interquartile range). HBV, hepatitis B virus; HCV, hepatitis C virus; SMI, skeletal muscle index; SATI, subcutaneous adipose tissue index; VATI, visceral adipose tissue index; CR, complete response; PR, partial response; SD, stable disease; PD, progressive disease.

**Table 2 cancers-15-04223-t002:** The pre-, combination-, and post-treatment of the enrolled patients.

	All Patients (*n* = 77)		
	Pre-Treatment	Combination Treatment	Post-Treatment
Any treatments	71 (92.2%)	49 (63.6%)	31 (40.3%)
Hepatectomy	35	1	0
RFA	20	4	1
TACE	57	43	19
Radiation therapy	18	11	7
Other chemotherapy	5	0	15

RFA, radiofrequency ablation; TACE, transcatheter arterial chemo embolization.

**Table 3 cancers-15-04223-t003:** Adverse events during chemotherapy.

	All Patients (*n* = 77)			
	Any Grade	Grade 1	Grade 2	Grade ≥ 3
Any symptoms	75 (97.4%)			24 (31.2%)
Appetite loss	45 (58.4%)	7 (9.1%)	36 (46.8%)	2 (2.6%)
Hand-foot syndrome	36 (46.8%)	13 (16.9%)	22 (28.6%)	1 (1.3%)
General fatigue	31 (40.3%)	3 (3.9%)	25 (32.5%)	3 (3.9%)
Hypertension	30 (39.0%)	2 (2.6%)	20 (26.0%)	8 (10.4%)
Diarrhea	18 (23.4%)	14 (18.2%)	4 (5.2%)	0
Proteinuria	15 (19.5%)	3 (3.9%)	8 (10.4%)	4 (5.2%)
Hypothyroidism	10 (13.0%)	0	10 (13.0%)	0
Hemorrhage	9 (11.7%)	1 (1.3%)	4 (5.2%)	4 (5.2%)
Liver dysfunction	8 (10.4%)	2 (2.6%)	3 (3.9%)	3 (3.9%)

**Table 4 cancers-15-04223-t004:** Analysis of prognostic factors that would affect OS using time-varying covariates in the Cox model.

Variable	Univariate Analysis		Mutivariate Analysis	
	HR (95%CI)	*p* Value	HR (95%CI)	*p* Value
Age (year)	0.992 (0.973–1.026)	0.952		
Sex (male vs. female)	1.007 (0.452–2.245)	0.986		
Presence of pre-treatment (yes vs. no)	2.480 (1.046–5.879)	0.039	2.995 (1.040–8.620)	0.042
AFP (10^3^ × ng/mL)	1.002 (1.001–1.002)	<0.001	1.002 (1.001–1.002)	0.002
PIVKA-II (10^3^ × mAU/mL)	1.006 (1.004–1.008)	<0.001	1.004 (1.002–1.006)	<0.001
ALBI score	3.564 (2.503–5.076)	<0.001	3.609 (2.342–5.562)	<0.001
SMI (cm^2^/m^2^)	0.904 (0.872–0.937)	<0.001	0.941 (0.906–0.977)	0.001
SATI (cm^2^/m^2^)	0.988 (0.976–1.001)	0.074	1.004 (0.990–1.017)	0.595
VATI (cm^2^/m^2^)	0.973 (0.959–0.988)	<0.001	0.990 (0.973–1.007)	0.263

OS, overall survival; HR, hazard ratio; CI, confidence interval; SMI, skeletal muscle index; SATI, subcutaneous adipose tissue index; VATI, visceral adipose tissue index.

## Data Availability

The data presented in this study are available upon request from the corresponding author.

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
