# Peer review of "Lenvatinib or Sorafenib Treatment Causing a Decrease in Skeletal Muscle Mass, an Independent Prognostic Factor in Hepatocellular Carcinoma: A Survival Analysis Using Time-Varying Covariates"

_cancers, 2023, doi:10.3390/cancers15174223_

Round 1
Reviewer 1 Report
The paper is interesting and overall well written. THe sample size is not large and this is an important limitation. Other limitation is the retrospective design of the study.
The authors should comment on the other definitions of sarcopenia that could be used in this field and their potential impact on the outcomes.
The authors should comment on the potential impact of skeletal muscle mass also on post-progression survival in these patients (cite the series PMID: 25085684)
The authors should comment also on the role of other diseases on the outcomes of these patients (cite PMID: 23845075)
Author Response
Response to Reviewer #1
We are pleased that in the overall comments this reviewer found our study is of interest. We also thank this reviewer’s constructive comments which were most helpful to improve our manuscript. We accordingly revised the manuscript as follows.
The paper is interesting and overall well written. The sample size is not large and this is an important limitation. Other limitation is the retrospective design of the study.
We fully recognized that small sample size and the retrospective study design are the main limitations in this study. We emphasized this point in the Discussion section (lines 276-277).
The authors should comment on the other definitions of sarcopenia that could be used in this field and their potential impact on the outcomes.
According to this suggestion, we referred to the definition of sarcopenia and the previous studies that demonstrated the relationship between sarcopenia and poor survival in patients with HCC and cirrhosis (see new Reference #5, #24, #25). Furthermore, we emphasized the possibility that patients with cirrhosis and HCC tend to lose their skeletal muscle, one of the prognostic factors, as the underlying liver diseases progressed (lines 235-243). These revisions enhance the quality of the paper. Thank you for your valuable suggestion.
The authors should comment on the potential impact of skeletal muscle mass also on post-progression survival in these patients (cite the series PMID: 25085684)
As you mentioned, advances in multidisciplinary treatment have made post-progression survival more important for HCC therapy in recent years. We referred to the possibility that sarcopenia could negatively affected post progression survival in patients with HCC with citing the suggested references (lines 243-247 and see new References #26, #27, #28). Thank you again for your important suggestion.
The authors should comment also on the role of other diseases on the outcomes of these patients (cite PMID: 23845075)
As you suggested, obesity and diabetes could negatively affect survival in patients with HCC (see new Reference #34). Insulin resistance and increased visceral fat volume are risk factors for HCC recurrence (see new References #20 and #35). Therefore, we added the sentence that proper nutrition and exercise therapies are important in combating obesity and diabetes, which increase the risk of HCC and worsen prognosis (lines 258-261). We believe this statement improves the paper. We thank your valuable comment again.
We thank this reviewer’s constructive comments which were most helpful to improve our manuscript.

Reviewer 2 Report
From a biostats and clinical epidemiology point of view, this research has been well planned, here are some suggestions for the Authors:
- more than time-dependent covariates, they are time-varying ones, do you agree? Please, refer to the classical definitions by Terry Therneau at Mayo Clinic
- line 114 “paired t-test” I do warmly suggest you to generally apply a non-parametric approach, in this case the Wilcoxon paired test, i.e.
- table 1 and everywhere, continuous covariates have to be reported as median/IQR
- table 2, the OS and PFS total number of events have to be reported, even to prove that the multivariate models are not overparametrized
- median follow-up info is lacking, better overall and by the two drugs
- KM curves, the y-axis report cumulative survivals more than rates
- figures 2 and 3, extremely informative! Anyway, an overall p-value for time series will be more correct (estimated by a mixed effects linear model for repeated measures)
- table 4, please add OS in the caption
- table 4, just before to comment it, we need to know the exact number of OS events (=dead pts)
- table 4, no PFS modeling too, why!? No age and gender estimation, why!? No previous treatment effect estimation, why!?
- table 5, I do suggest to discard it, only fews comments in the text will be sufficient
Author Response
Response to Reviewer #2
We are pleased that in the overall comments this reviewer found our study is of interest. We also thank this reviewer’s constructive comments which were most helpful to improve our manuscript. We accordingly revised the manuscript as follows.
- more than time-dependent covariates, they are time-varying ones, do you agree? Please, refer to the classical definitions by Terry Therneau at Mayo Clinic
According to this suggestion, we changed all the word ‘time-dependent covariates’ into ‘time-varying covariates.’ Thank you for pointing out the correct statement.
- line 114 “paired t-test” I do warmly suggest you to generally apply a non-parametric approach, in this case the Wilcoxon paired test, i.e.
According to this suggestion, we reanalyzed using a Wilcoxon paired test (lines 114) and rewrote the corresponding results (lines 166, 177-178 and Figure 2 and 3).
- table 1 and everywhere, continuous covariates have to be reported as median/IQR
According to this suggestion, we revised Table 1 and Table S2 (Table5).
- table 2, the OS and PFS total number of events have to be reported, even to prove that the multivariate models are not overparametrized
- median follow-up info is lacking, better overall and by the two drugs
- table 4, just before to comment it, we need to know the exact number of OS events (=dead pts)
We thank your crucial comments. During the observation period, 68 patients experienced PD and 53 died in the present study. The median follow-up period for all patients, those treated with LEN, and SOR were 20.9, 15.6, and 21.8 months, respectively. We added this information in the Materials and Methods section (lines 143-146).
- KM curves, the y-axis report cumulative survivals more than rates
According to this suggestion, we revised Figure 1.
- figures 2 and 3, extremely informative! Anyway, an overall p-value for time series will be more correct (estimated by a mixed effects linear model for repeated measures)
According to this suggestion, we analyzed overall p-values for time series for SMI, SATI, VATI, AFP, PIVKA-II, and ALBI score. All the parameters other than AFP significantly turned out to changed over time. We added this information in the Statistical analysis subsection (lines 114-115) and Results section (lines 179-180).
- table 4, please add OS in the caption
According to this suggestion, we added OS in the caption of Table 4.
- table 4, no PFS modeling too, why!? No age and gender estimation, why!? No previous treatment effect estimation, why!?
According to this suggestion, we estimated the impact of age, gender, and the presence of pretreatment on OS. Among them, the presence of pre-treatment was chosen as an independent factor that would affect OS. We revised Table 4 and the description in the text (lines 31-33 and lines 190-193). We also evaluated factors that would affect PFS in the same manner (Table S1). AFP (HR: 1.002, 95% CI: 1.001–1.003, p = 0.007), PIVKA-II (HR: 1.004, 95% CI: 1.001–1.008, p = 0.012) and ALBI score (HR: 1.676, 95% CI: 1.145–2.452, p = 0.008) were independent factors that would affect PFS (lines 194-198).
- table 5, I do suggest to discard it, only fews comments in the text will be sufficient
According to this suggestion, we omitted Table 5 in the text. However, we think that the result of Table 5 is very informative. It means that we cannot predict the rapid depletion of skeletal muscle judging from drugs (LEN or SOR), AEs, liver functional reserve or clinical tumor stage at the introduction, and therapeutic effect and that patients who experienced it had much poorer survival than those who did not. Furthermore, it implies that there are other unknown mechanisms that associate LEN/SOR treatment with the depletion of skeletal muscle. Thus, we decided to leave this table as Table S2.
We thank this reviewer’s constructive comments which were most helpful to improve our manuscript.

Round 2
Reviewer 1 Report
The revised version of the paper is OK. Thank you!
Reviewer 2 Report
The Authors were able to notably improve their manuscript, that deserves to be published